# The Anatomical Conditions of the Alveolar Process of the Anterior Maxilla in Terms of Immediate Implantation—Radiological Retrospective Case Series Study

**DOI:** 10.3390/jcm10081688

**Published:** 2021-04-14

**Authors:** Piotr Wychowański, Anna Starzyńska, Martyna Osiak, Jan Kowalski, Barbara Alicja Jereczek-Fossa, Blanka Seklecka, Tadeusz Morawiec, Paulina Adamska, Jarosław Woliński

**Affiliations:** 1Department of Oral Surgery, Medical University of Warsaw, 6 Binieckiego Street, 02-097 Warsaw, Poland; piotrwychowanski@wychowanski.pl (P.W.); martynaosiak88@gmail.com (M.O.); 2Department of Oral Surgery, Medical University of Gdańsk, 7 Dębinki Street, 80-211 Gdańsk, Poland; paulina.palasz@gumed.edu.pl; 3Department of Periodontology and Oral Diseases, Medical University of Warsaw, 6 Binieckiego Street, 02-097 Warsaw, Poland; jkowalski@wum.edu.pl; 4Department of Oncology and Hemato-Oncology, University of Milan, 7 Festa del Perdono Street, 20-112 Milan, Italy; barbara.jereczek@ieo.it; 5Division of Radiotherapy, IEO European Institute of Oncology, IRCCS, 435 Ripamonti Street, 20-141 Milan, Italy; 6Early Clinical Trials Unit, University Clinical Centre, 17 Smoluchowskiego Street, 80-214 Gdańsk, Poland; bseklecka@uck.gda.pl; 7Department of Oncology and Radiotherapy, Medical University of Gdansk, 3a M. Skłodowskiej-Curie Street, 80-210 Gdańsk, Poland; 8Department of Oral Surgery Silesian Medical University, 17 Plac Akademicki Street, 41-902 Bytom, Poland; tmorawiec@sum.edu.pl; 9Department of Animal Physiology, The Kielanowski Institute of Animal Physiology and Nutrition, Polish Academy of Sciences, 3 Instytucka Street, 05-110 Jabłonna, Poland; j.wolinski@ifzz.pan.pl

**Keywords:** dental implants, immediate implantation, flapless implants, predictable implantation, cone-beam computer tomography (CBCT), alveolar bone shape, bone anatomy, classification, aesthetic region of maxilla

## Abstract

The feasibility and the level of difficulty of immediate flapless implantation depend largely on the residual alveolar bone. The purpose of the study was to determine how often immediate flapless implantation in the anterior maxilla is feasible and assess the difficulty level using cone-beam computed tomography (CBCT) scans. A radiological retrospective case series study was conducted. In total, 1200 CBCT scans from 300 consecutive patients were analyzed with dedicated planning software. Immediate flapless implants were possible in 78.33% of cases. Drilling direction was either through the apex or the palatal slope. **Bimodal** was conducted in 9% of the cases; only through the apex in 13.08% of the cases and in 56.25% only in the slope. In 21.67%, immediate flapless implants were excluded. The feasibility and degree of difficulty differed statistically to the disadvantage of the lateral incisors compared to the central incisors. Drilling direction caused that BASE classification reflects the difficulty level of immediate implantation. CBCT is a helpful diagnostic tool for assessing the feasibility of immediate flapless implants due to the residual bone shape and volume. BASE classification helps to determine a challenge level that may also facilitate communication and result in comparison. The alveolar bone condition allows for immediate flapless implants in most cases in the aesthetic region of the maxilla, but they should be performed by an experienced specialist with regard to the bone and soft tissue quality.

## 1. Introduction

Dental implants are a recognized, safe, and long-lasting method of replacing missing teeth, appreciated by patients and leading to an increase in oral health quality of life [1,2,3]. Intensive development of scientific research concerning novel materials used for the production of dental implants and the modifications of their surfaces is currently being observed. The aim of this observation is to accelerate the increase of osteointegration with the patient’s bone [3,4]. The research activity, which is based on the strength and durability of osteointegration, results from the patient’s expectation to expand qualifying indications for dental implantology. The improvement of osteointegration may enable dental implants for patients previously refused due to general health conditions or habits, such as smoking [5,6,7,8]. Enhanced osseointegration may be important in the case of inserting implants in difficult bone conditions resulting from post-extraction bone atrophy of the jaws [9]. If bone and soft tissue deficiencies are found, it is necessary to use short implants, mini-implants, or zygomatic implants [10,11,12].

The research on the novel osteoinductive or osteoconductive bone substitute materials and their application methods in bone and soft tissue augmentation may arise a pragmatic alternative for these techniques [13,14]. Alveolar bone atrophy after a tooth extraction is much less revealed when immediate implantation is performed [15,16]. The immediate implant concept was described in the 1970s [17] and became further pervasive in the 1980s [18,19]. It preserves the buccal bone plate, and it reduces the number of surgeries, overall treatment time, and cost. It also increases patient acceptance for implant treatment. Immediate post-extraction implant placement makes benefits for all the bone available and soft tissues, which tend to decrease in volume for delayed or late implants. Furthermore, immediate implants have already demonstrated similar success rates to staged implants [20,21,22,23].

The primary implant stability is the result of implant features, such as implant body shape, thread design, and surface topography, as well as surgical technique and the patient’s hard tissue conditions [24,25,26]. The factors influencing primary stability with other clinical factors, such as implant neck shape and position, provisional and definitive prothesis, are the main prognostic factors for achieving secondary stabilization and the long-term success of the implant treatment [27].

High-primary implant stability depends on optimal positioning of the implant and its suitable design in available bone. The depth, angulation, and distance from the adjacent teeth or implants determine the future stability of the peri-implant bone and soft tissues and thus the overall success and implant survival [25]. The clinical effectiveness of the positioning of the implants in the bone and the assessment of their final location can be achieved both by using dental templates or free-hand operations [10,28,29].

In this respect, cone-beam computed tomography (CBCT), with dedicated software, is an excellent tool to evaluate the available bone and plan for optimal implant shape and position. A variety of implant shapes and thread designs enable optimal accommodation of the implant to the available bone, both compact and spongious, to achieve sufficient primary stability [10,30,31].

Flapless immediate implants are the most challenging; however, they are also the most desirable to preserve blood supply and the buccal plate [31,32,33,34]. The post-extraction socket orifice is the only entry, and therefore, the study of the most commonly encountered patterns of bone anatomy in this maxillary region is desirable. In the available literature, there are a few publications on the shape and quality of bone of the alveolar process, and there is a great need to describe the alveolar bone typical anatomy directly after tooth extraction in terms of the possibility of immediate implantation [35,36,37,38,39,40,41,42,43,44]. In an animal model, immediately after a tooth extraction, the molecular and histologic study by Marconcini et al. reported that myofibroblasts were up-regulated in the first 15-days of the healing period; thus, the excessive tension of these cells during the epithelial closure of the socket may be managed by the use of collagen, an immediate implant placement, or a provisional prosthesis [45].

The primary objective of the study was to assess if CBCT scans can be a sufficient tool to decide for immediate implants in the aesthetic zone. The secondary objective of the study was to analyze the statistics of different bone profiles in the aesthetic zone in patients coming for immediate implants. The third objective was to assess how many consecutive patients could receive immediate implants and how difficult it may be for a surgeon and restorative dentists to provide safe treatment with a predictable outcome.

## 2. Material and Methods

A radiological retrospective case series study was conducted with the consent of the bioethics committee no. AKBE/221/2020 issued by the Bioethical Commission at the Medical University of Warsaw.

Three hundred patients were enrolled in the study: 164 males and 136 females in the age from 20 to 64 years (mean age 51.2 years). The patients referred in the years 2013–2020 to our clinic were generally healthy and were diagnosed for tooth extraction in the maxillary aesthetic region. The inclusion criteria for the study were that patients prior to treatment had to have all premolars, canines, and incisors in the maxilla and a stable lateral occlusal support. The exclusion criteria were prior dental implant treatments, surgical interventions, trauma or any other pathology causing bone remodeling in the front maxilla diagnosed on radiograms or stated in the patient’s medical records. Pregnant patients were also excluded.

The study was conducted based on cone-beam computer tomography scans of the central and lateral incisors, canines, and first and second premolars. The scans were made with the Kodak 9000C 3D (Kodak Dental Systems, Atlanta, GA, USA). Measurements were carried out using a 50 mm × 37 mm imaging field providing a resolution of 76 μm. The scan analysis was performed by three independent researchers using the Kodak Dental Imaging Software 6.12.32 version. Names of the patients, age, gender, and their treatment history were unknown for the researchers. The order of the scans was random. The final reported results were the mean values of all obtained scans from the patients enrolled in the study. In total, the researchers analyzed 1200 cross-section (perpendicular) scans of individual teeth and the surrounding bone, 300 of each were central and lateral incisors on both the right and left sides. The first step of the study was to measure bone on the cross-section scans before extraction, as shown in Figure 1.

In the second stage, using dedicated software, the implant placement was planned in a way that would not further compromise the socket bone walls. Furthermore, recent recommendations for proper 3D implant positioning were applied with respect to biological width. Minimum half of the implant circumference at any level should be embedded in native bone. The implant phantoms used for planning were from 3 mm to 4.5 mm in diameter and from 10 mm to 16 mm in length in order to achieve the most bone to implant contact, and the implant angulation was no more than 25 degrees to the prosthetic crown axis, as shown at Figure 2 [14,25,26,32].

The sample size was estimated using G*Power software, version 3.1.9.4, for a one-tailed t-test at *α* = 0.05 with 95% power, assuming d (effect size) = 3, for the study groups. The recommended total sample size was 254 observations. The effect size 3 was reached assuming minimal *α* = 24.5 and maximal SD = 96.17. All data were expressed as the number of observations per group (n and %). The chi-square distribution or Fisher’s exact test was used to compare the differences between the groups. The critical values of the chi-square (*X*^2^) distribution were 7.815. If possible, the significance test was 2-tailed and conducted at a 0.05 level of significance.

## 3. Results

A total of 300 CBCT scans were analyzed. The alveolar bone condition in the areas of both lateral and both central incisors was assessed on each scan. In total, the bone was measured and assessed in terms of the feasibility of immediate implants in 1200 locations (perpendicular scans). Exercises for implant planning led to the identification of four repeating patterns of bone anatomy around teeth to be extracted. These patterns were either excluded or included for immediate implants and defined the drilling direction (Figure 3, Table 1). Immediate flapless implants were possible in 940 amongst 1200 diagnosed implant sites (78.33% of all cases). The drilling direction was either through the apex (along the long axis of the socket) or the palatal slope (oblique to the long axis of the socket) in 108 implant sites, 157 implant sites could only be through the apex and 675 implant sites could only be through the slope of the palate. In 260 implant sites (21.67% of all cases), immediate implants were not possible.

Thus, the **BASE** classification was made. The BASE classification consists of four modalities of the alveolar bone shape, which is crucial for immediate flapless implant insertion planning and longtime success.

In the **B**imodal group (9% of all cases), implant bed preparation could either start at the **a**pex of the socket along its axis or go sideways to the **s**lope of the palate. Therefore, we named it **Ba**—for **B**imodal **a**pex and **Bs**—for **B**imodal **s**lope. In 13.08% of all studied cases, the bone anatomy allowed only for preparation through the **a**pex along the socket long axis, and therefore, such ridge pattern we, later after full analysis, named **A**—for **A**pex. In more than half of the implant sites (56.25% of all cases), the only way to achieve primary implant stability was to go sideways into the **s**lope of the palate, and this kind of anatomy pattern we named **S**—for **S**lope. In more than one-fifth of examined scans (21.67% of all cases), immediate implants had to be excluded because bone anatomy would not allow for sufficient embedment of an implant in the native bone to enable predictable primary implant stability. We named such bone pattern **E**—for **E**xcluding.

Putting the four ridge patterns together in order of growing complexity we arrived at the **BASE** acronym, which seems to practically express drilling guidelines and difficulty level for immediate flapless implants in the aesthetic region, as shown in Figure 3.

Further analyses revealed the differences between the local bone conditions of the maxillary alveolar process in the area of the central and lateral incisors in terms of planning immediate implantation. Immediate implant placement was feasible in 538 cases in the central incisor region, while in the lateral incisor region in only 402 cases. The most common variation of the alveolar bone as compared with all possible bone modalities or as compared with the alveolar bone modalities enabling immediate implant placements was the implant cradle preparation starting on the slope of the fresh alveolar socket. The differences were statistically significant (Table 1, Figure 4). There were no differences when comparing the bone conditions of the maxillary alveolar process on the left side to the right side, both in the central and lateral incisor areas (Table 1, Figure 5).

## 4. Discussion

Flapless dental implantation is a current research topic. Flapless dental implantation provides patients with less traumatization of tissues, reduction of postoperative reactions, a faster recovery period and return to full health and social functionality. There is a wide diversity of flapless implantation techniques. The flapless implants are provided in various regions of a jaw, in various time algorithms, or in guided or free-hand procedures. All these variables have a significant impact on the conditions and goals of the procedure and should be recognized and taken into account at the planning stage [10,15,20,21,22,29,30,32,46]. Current literature describes bony conditions for immediate implants [20,31,35,36,37,38,39,40,41,42,43,44,47]. Some of them focus primarily on internal bone structure and bone density [36,39,41,42]. The classification of the alveolar bone conditions proposed by Papadimitriou et al. is applicable to edentulous arches [43]. Smith and Tarnow proposed three stages classification for immediate implants in the molar area [38]. The classification by Buser et al. refers to the time elapsed from tooth extraction to implantation [20]. Corbella and Salama propose strategies for bone defects and soft tissue management around the immediate implants [37,40]. SAC classification [44] seems to be the most complete; however, it does not explore implant positioning in the bone outside the extraction socket—namely in the slope or the palate (straightforward—S; advanced—A; complex—C). Kan et al. analyzed 600 scans of maxillary incisors and canines, but there are still some limitations concerning implant positioning to the alveolar socket [35].

This study was designed to explore the statistical opportunity for immediate flapless implants not only in the alveolar socket but also as much palatally as prosthetically manageable. The results may bring practical guidelines for planning the implant position and drilling directions in a straightforward classification. In our study, local bone conditions allow for immediate implants in more than 78% of cases (B+A+S modalities), which is very frequent. In 22.80% of the cases, it could be a comparatively easy procedure due to the longitudinal axis of drilling for the implant bed and the sharp starting point located in the apex of the socket (class B and A). For more than half of the cases (56.25%—Class S), immediate implants were possible but more challenging due to the inaccurate starting point at the palatal slope of the socket and inclined axis of drilling for the implant bed. The drilling starting point is not very sharp and easily visible if performing flapless implantation. Moreover, the starting point is frequently covered by dense cortical bone and may cause the drill to slip, and consequently, the drilling axes are changed. It should be considered to perform these cases by an experienced surgeon. Guided surgery and surgical templates should be recommended in these cases [30]. In 21.67% of cases (260 implant sites—Class E), immediate implants were not possible in our study. That means that almost one out of three patients desired immediate implantation in the aesthetic region and had to undergo staged therapy. The post-extraction bone remodeling process consumes up to 20% of the native bone vertically and up to 60% horizontally within the first 6 months. These reasons urge researchers to fully explore the potential for immediate implants [15,18,19,20,33]. The results of flapless immediate implants in anterior maxilla differ by authors with survival and aesthetic success ratio. There are some contradictory recommendations. It could be due to the combined assessment and comparisons of cases with strongly different local bone and soft tissue conditions [15,20,21,22,32,33,34,37]. It allows classifying the immediate implantations cases into more homogenous groups. This study was designed to describe the basic repeated patterns of the alveolar process in the anterior maxilla founded by the surgeon at the time of the immediate flapless implantation. It could give practical drilling directions, not limited to the alveolar socket, to achieve sufficient primary stability. BASE classification, which is the result of this investigation, defines the level of difficulty of procedures with straightforward (easy) drilling in classes Ba and A and more challenging classes Bs and S for both the surgeons and restorative dentists. Class E characterizes the bone anatomy, which does not provide predictable primary stability for an immediate implant and calls for staged, planned restorative surgeries. This study showed statistically significant differences in the shape of the alveolar bone in the anterior maxilla in the region of the central incisor as compared to the lateral incisor. Immediate implants in the area of the lateral maxillary incisor are procedures of a higher degree of difficulty than in the area of the central incisor. More often, one has to abandon simultaneous procedures in favor of multi-stage treatment plans. However, the limitation of the study is the quality of the CBCT and software used for planning the immediate implantation. Our observations are consistent with the results of the research of Zhang W. et al. [30]. While examining the anatomy of the maxillary alveolar bone, they found that the lateral incisor had a significantly smaller alveolar width than the other anterior teeth, and buccal undercut occurs almost twice more frequently and is deeper at the lateral incisor compared to the central incisor. The advantages of this study are predictability, simple planning based on the CBCT study, and following the study protocol. Thanks to the classification, it is possible to divide patients in whom immediate implantation can be performed or not. However, it should be remembered that the volume and shape of the alveolar bone after tooth extraction, necessary for the correct three-dimensional positioning of the immediate implant, is not the only factor determining the success of implant treatment. The other variables that must be taken into account when planning immediate flapless implantations are bone density, the ratio of the cortical to cancellous bone, buccal bone plate thickness, as well as soft tissue biotype. The first two factors determine primary implant stabilization, while the other two are essential in biological reactions resulting in the secondary implant stabilization and long-term soft tissue condition. It is the result of successful osseointegration. The shape, material and surface of the implant used are important, as well as the technique of implant cradle preparation [4,20,26,32,46,47,48]. The variability of local soft and hard tissue conditions found in the cases of immediate implant insertion in the aesthetic region of the maxilla may result in different levels of difficulty and predictability of the treatment. Most of the time, bone anatomy in the front maxillary region is challenging for proper implant positioning and anchoring. This may be unheeded, especially when flapless surgery is conducted. Moreover, the thin buccal plate bone can be further compromised by apical lesions. Thin biotype of soft tissues is also more common. Therefore, minimally invasive flapless surgery calls for thorough pre-operative clinic diagnostics supported with cone-beam CT scan evaluation [25,31,34]. The potential risks of radiation exposure with CBCT seem well balanced by the benefits observed in our study.

## 5. Conclusions

Cone-beam computed tomography and dedicated software are very helpful diagnostic tools to assess the feasibility of immediate flapless dental implants in the anterior maxilla due to the residual bone shape and volume. BASE classification helps to determine the drilling direction and level of challenge for surgeons and restorative dentists. It may also facilitate communication between researchers and enable more precise result comparison in clinical studies. The alveolar bone condition allows for immediate flapless implants in most cases in the aesthetic region of the maxilla, but they should be performed by an experienced specialist with regard to the bone and soft tissue quality.

## Figures and Tables

**Figure 1 jcm-10-01688-f001:**
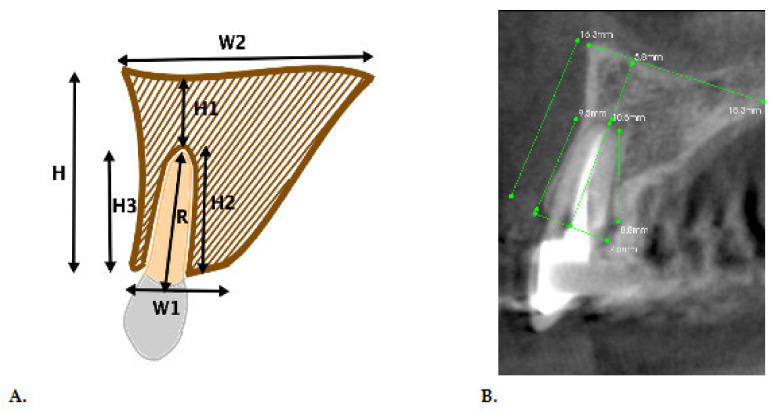
Measurements of the alveolar bone on cross-section cone-beam computed tomography (CBCT) scans in the anterior maxilla. (**A**). Schematic drawing; (**B**). Radiological measurements (H—Total height of the alveolar process; H1—The height of the bone from the apical part of the alveoli to the bottom of the nose or the maxillary sinus; H2—The height of the palatal plate of the alveoli; H3—The height of the buccal plate of the alveoli; R—The root length of the tooth; W1—The width of the bone at the level of the alveoli aperture; W2—The width of the bone at the basis).

**Figure 2 jcm-10-01688-f002:**
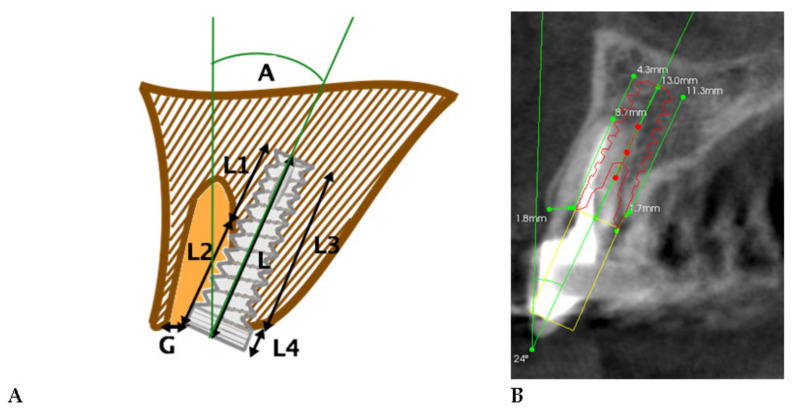
Measurements of the planned implant relationship to the alveolar process on cross-sectional CBCT scans in the anterior maxilla. (**A**). Scheme; (**B**). Radiological example of the taken measurements (L—Total implant length; L1- Implant length submerged in the bone on the buccal side; L2—Implant length protruding from the bone on the buccal side; L3—Implant length submerged in the bone on the palatal side; L4—Implant length protruding from the bone on the palatal side; G—Gap between the buccal surface of the implant and the bone buccal plate).

**Figure 3 jcm-10-01688-f003:**
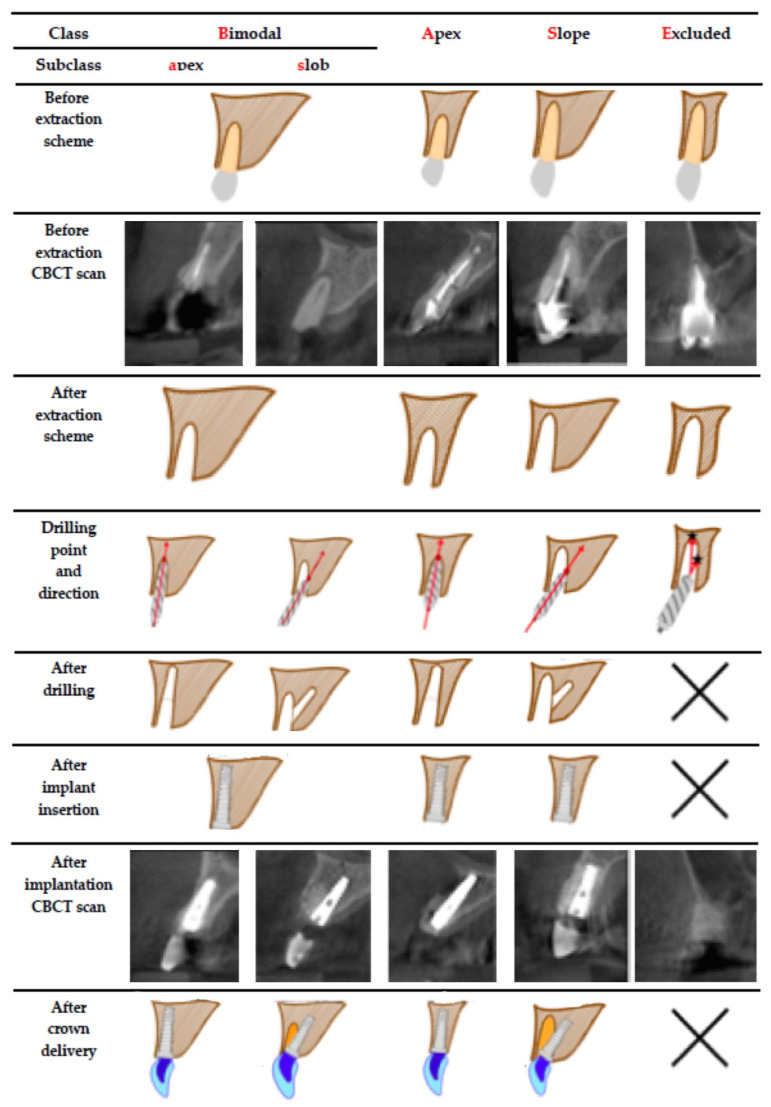
BASE—drilling guidelines for immediate flapless implants in the maxillary aesthetic region.

**Figure 4 jcm-10-01688-f004:**
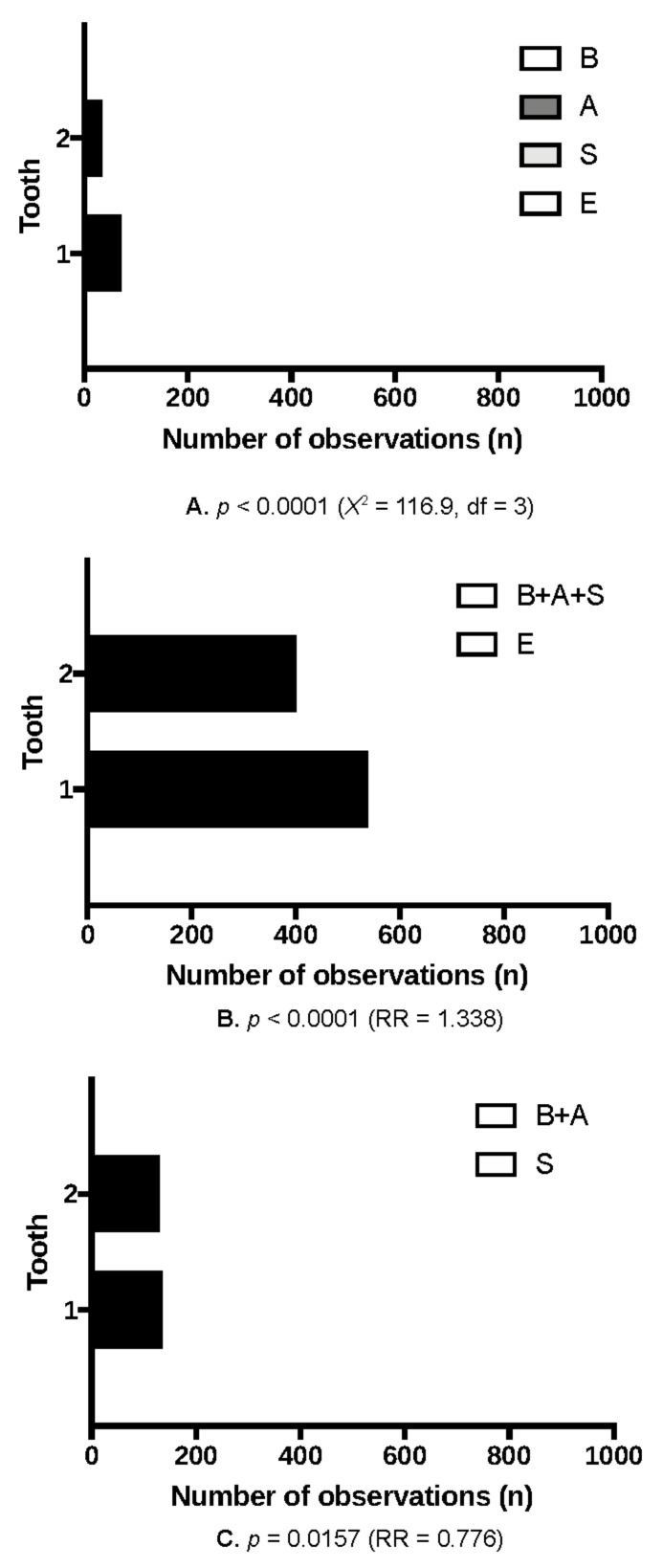
The most often alveolar bone modalities of the anterior maxilla depending on the tooth type. BASE classification: (**A**). The frequency of all modalities; (**B**). The feasibility of immediate implantation versus cases excluded; (**C**). The difficulty level of immediate implantation (1—all central incisors, 2—all lateral incisors).

**Figure 5 jcm-10-01688-f005:**
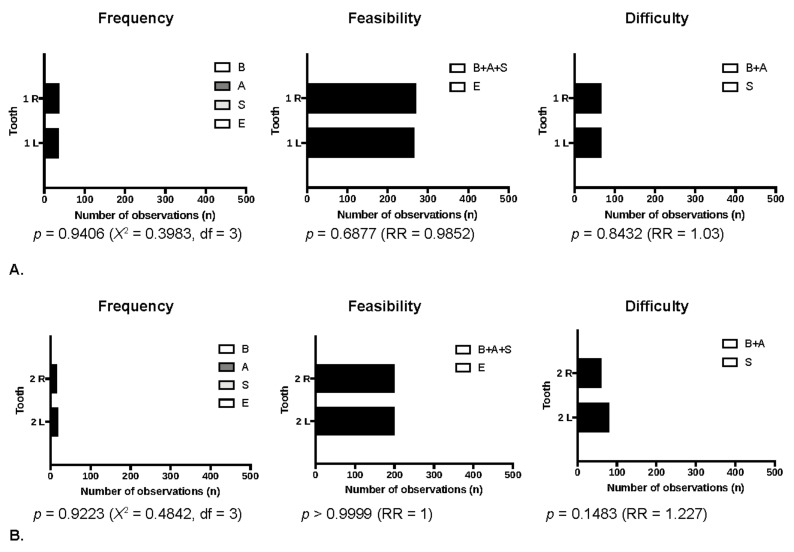
The dental arch side dependent on bone condition comparison. BASE Classification: (**A**). Central incisors; (**B**). Lateral incisors (1 R—right central incisors; 1 L—left central incisors; 2 R—right lateral incisors, 2 L—left lateral incisors).

**Table 1 jcm-10-01688-t001:** Occurrence of individual anatomical classes of the maxillary alveolar process in the anterior region—BASE classification (1—all central incisors; 2—all lateral incisors; L—left side of the dental arch; R—right side of the dental arch).

Class	Total				Bimodal				Apex				Slope				Excluded				B+A+E (Immediate Implantation Possible)			
Percentage in total	100				9				13.08				56.25				21.67				78.33			
Number of implants	1200				108				157				675				260				940			
Tooth type	1		2		1		2		1		2		1		2		1		2		1		2	
Number of implants	600		600		73		35		62		95		403		272		62		198		538		402	
Percentage in class	50		50		67.60		32.40		39.49		60.51		59.70		40.30		23.85		76.15		57.23		42.77	
Percentage of all sides	50		60		6.08		2.92		5.16		7.92		33.59		22.66		5.17		16.50		44.83		33.50	
Tooth side	L		R		L		R		L		R		L		R		L		R		L		R	
Number of implants	600		600		53		55		76		81		343		332		128		132		472		468	
Percentage of all sides	50		50		49.08		50.92		48.41		51.59		50.81		49.19		49.23		50.77		50.21		49.79	
Tooth type and side	12	11	21	22	12	11	21	22	12	11	21	22	12	11	21	22	12	11	21	22	12	11	21	22
Number of implants	300	300	300	300	16	37	36	19	46	30	32	49	139	204	199	133	99	29	33	99	201	271	267	201
Percentage of all sides	25	25	25	25	1.34	3.08	3	1.58	3.83	2.50	2.67	4.08	11.58	17	16.58	11.09	8.25	2.42	2.75	8.25	16.75	22.58	22.25	16.75

## Data Availability

The data presented in this study are available on request from the corresponding author.

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
