# Peer review of "The Anatomical Conditions of the Alveolar Process of the Anterior Maxilla in Terms of Immediate Implantation—Radiological Retrospective Case Series Study"

_jcm, 2021, doi:10.3390/jcm10081688_

Round 1
Reviewer 1 Report
This paper is an interesting clinical study that has focused on CBCT accuracy with implant treatment. These studies were significant for understanding of implant treatment. However, there are a few concerns need to be addressed.
The results of this study are greatly influenced by the accuracy of the CBCT machine. Then, I think that these results will change depending on which manufacturer's CBCT is used. What do the authors think? This seems to be the most important part of this research.
Furthermore, although the evaluation on the sagittal plane is solid, it seems necessary to evaluate the difficulty level on the frontal plane and the horizontal plane.
Author Response
Dear Reviewer,
Thank you for your analysis and the remarks. We revised the structure of the article in accordance with your suggestions.
In detail:
1. The results of this study are greatly influenced by the accuracy of the CBCT machine. Then, I think that these results will change depending on which manufacturer's CBCT is used. What do the authors think? This seems to be the most important part of this research.
Response 1: The type of CBCT machine used in the study may play a role, especially the voxel resolution is very important for the accuracy of measurement and planning. Some CBCT machines have the option to switch the scan field of view and the resolution of the scans. The higher field of view you use, the lover resolution you obtain. In our study we try to use small image areas in order to obtain high resolution and accuracy. We assumed that a resolution of 76 μm is sufficient for the accuracy of the study, as it is one of the best actually available ones. We used the 50x37 mm field of view. It was sufficient to include all examined teeth in one scan. We add new information in section: material and methods (page 3; lines 143-144): Measurements were carried out using 50x37 mm imaging field providing resolution 76 μm.
2. Furthermore, although the evaluation on the sagittal plane is solid, it seems necessary to evaluate the difficulty level on the frontal plane and the horizontal plane.
Response 2: The sagittal plane is the starting point in the process of case selection and difficulty level assessment, especially in the case of immediate implant placement. The frontal plane and horizontal plane are the two that are taken in account as well, just to respect the principles of biological width of the tissue. It could be of most importance in the cases of anodontia or postextraction bone deficiencies. In our research we examined patients scans before existing tooth extractions, so we do not observed deficiencies in horizontal and sagittal planes and we just adapted implant planning to the existing bone conditions. We plan to continue the research taking into account variables you suggest.
Best regards

Reviewer 2 Report
Dear authors this paper is suitable for publication according to me for Journal of Clinical Medicine because it talks about an hot topic in dentistry and implantology, interesting in research and clinical activities. I agree with authors about the thematic and the approach of the article and surgery. I only want to suggest some papers needed to be cited that you may have lost during your writing and it could be necessary to cite them to perform an optimum article. And this is a very great article. I will judge it very positive to the editor.
LINE 1-3: This first sentence is true, but I would cite this following article instead of the number 1 because is more recent: 2018 vs 2015; and it has a longer follow-up.
Cosola S, Marconcini S, Giammarinaro E, Poli GL, Covani U, Barone A. Oral health-related quality of life and clinical outcomes of immediately or delayed loaded implants in the rehabilitation of edentulous jaws: a retrospective comparative study. Minerva Stomatol. 2018 Oct;67(5):189-195. doi: 10.23736/S0026-4970.
LINE 70-73: This sentence is fine, but at the end of it I suggest to add also the following citation because it is an overview on this topic.
De Santis D, Sinigaglia S, Pancera P, et all. An overview of socket preservation. J Biol Regul Homeost Agents. 2019 Jan-Feb;33(Suppl 1 [integer]):55-59.
LINE 74-75: Also here, I agree with the sentence but it is a very strong sentence so I suggest to add one other citation to support it. The following:
Arora, H.; Khzam, N.; Roberts, D.; Bruce, W.L.; Ivanovski, S. Immediate implant placement and restoration in the anterior maxilla: Tissue dimensional changes after 2-5 year follow up. Clin. Implant Dent. Relat. Res. 2017, 19, 694–702.
LINE 84-89: I would to rewrite this sentence in this way. "The primary implant stability is the result of implant features such as implant body shape, thread design and surface topography, as well as surgical technique and patients hard tissue conditions [22, 23]. The factors influencing primary stability with other clinical factors such as implant neck shape and position, provvisional and definitive prothesis are main prognostic factor for achieving secondary stabilization and the long-term success of implant treatment. [new citation as follow].
Cosola S, Marconcini S, Boccuzzi M, Menchini Fabris GB, Covani U, Peñarrocha-Diago M, Peñarrocha-Oltra D. Radiological Outcomes of Bone-Level and Tissue-Level Dental Implants: Systematic Review. Int J Environ Res Public Health. 2020 Sep 22;17(18):6920. doi: 10.3390/ijerph17186920.
LINE 94-96: Also here, I agree with the sentence but it is a very strong sentence so I suggest to add one other citation to support it. The following:
Cosola S, Toti P, Peñarrocha-Diago M, Covani U, Brevi BC, Peñarrocha-Oltra D. Standardization of three-dimensional pose of cylindrical implants from intraoral radiographs: a preliminary study. BMC Oral Health. 2021 Mar 6;21(1):100. doi: 10.1186/s12903-021-01448-9.
LINE 109-11: Talking about immediately after tooth extraction I suggest you to cite also this novel work about a new role of myofibroblast. I suggest this sentece.
"In animal model, immediately after tooth extraction, the molecular and histologic study by Marconcini and colleagues reported that myofibroblasts were up-regulated in the first 15-day healing period, thus the excessive tension of these cells during the epithelial closure of the socket may be managed by the use of collagene, an immediate implant placement or a provvisional prothesis."
In this case, this is the citation:
Marconcini S, Denaro M, Cosola S, Gabriele M, Toti P, Mijiritsky E, Proietti A, Basolo F, Giammarinaro E, Covani U. Myofibroblast Gene Expression Profile after Tooth Extraction in the Rabbit. Materials (Basel). 2019 Nov 9;12(22):3697. doi: 10.3390/ma12223697.
DISCUSSION
You have to highligh the limitations of the study and possible future improvments. It is important for the Scientific Soundness of your paper.
Author Response
Dear Reviewer,
Thank you for your analysis and the remarks. We revised the structure of the article in accordance with your suggestions.
In detail:
1. (x) English language and style are fine/minor spell check required
Response 1: Thank you for the revision. We took all the comments into consideration. In consonance with your suggestion, a professional native speaker checked and edited the manuscript.
2. LINE 1-3: This first sentence is true, but I would cite this following article instead of the number 1 because is more recent: 2018 vs 2015; and it has a longer follow-up.
Cosola S, Marconcini S, Giammarinaro E, Poli GL, Covani U, Barone A. Oral health-related quality of life and clinical outcomes of immediately or delayed loaded implants in the rehabilitation of edentulous jaws: a retrospective comparative study. Minerva Stomatol. 2018 Oct;67(5):189-195. doi: 10.23736/S0026-4970.
Response 2: The correction in the introduction was made. (page 2, line 55)
3. LINE 70-73: This sentence is fine, but at the end of it I suggest to add also the following citation because it is an overview on this topic.
De Santis D, Sinigaglia S, Pancera P, et all. An overview of socket preservation. J Biol Regul Homeost Agents. 2019 Jan-Feb;33(Suppl 1 [integer]):55-59.
Response 3: The correction in the introduction was made. (page 2, lines 80-81)
4. LINE 74-75: Also here, I agree with the sentence but it is a very strong sentence so I suggest to add one other citation to support it. The following:
Arora, H.; Khzam, N.; Roberts, D.; Bruce, W.L.; Ivanovski, S. Immediate implant placement and restoration in the anterior maxilla: Tissue dimensional changes after 2-5 year follow up. Clin. Implant Dent. Relat. Res. 2017, 19, 694–702.
Response 4: The correction in the introduction was made. (page 2, lines 73)
5. LINE 84-89: I would to rewrite this sentence in this way. "The primary implant stability is the result of implant features such as implant body shape, thread design and surface topography, as well as surgical technique and patients hard tissue conditions [22, 23]. The factors influencing primary stability with other clinical factors such as implant neck shape and position, provvisional and definitive prothesis are main prognostic factor for achieving secondary stabilization and the long-term success of implant treatment. [new citation as follow].
Cosola S, Marconcini S, Boccuzzi M, Menchini Fabris GB, Covani U, Peñarrocha-Diago M, Peñarrocha-Oltra D. Radiological Outcomes of Bone-Level and Tissue-Level Dental Implants: Systematic Review. Int J Environ Res Public Health. 2020 Sep 22;17(18):6920. doi: 10.3390/ijerph17186920.
Response 5: The correction in the introduction was made. (page 2, lines 82-88)
6. LINE 94-96: Also here, I agree with the sentence but it is a very strong sentence so I suggest to add one other citation to support it. The following:
Cosola S, Toti P, Peñarrocha-Diago M, Covani U, Brevi BC, Peñarrocha-Oltra D. Standardization of three-dimensional pose of cylindrical implants from intraoral radiographs: a preliminary study. BMC Oral Health. 2021 Mar 6;21(1):100. doi: 10.1186/s12903-021-01448-9.
Response 6: The correction in the introduction was made. (page 2, lines 96)
7. LINE 109-11: Talking about immediately after tooth extraction I suggest you to cite also this novel work about a new role of myofibroblast. I suggest this sentece.
"In animal model, immediately after tooth extraction, the molecular and histologic study by Marconcini and colleagues reported that myofibroblasts were up-regulated in the first 15-day healing period, thus the excessive tension of these cells during the epithelial closure of the socket may be managed by the use of collagene, an immediate implant placement or a provvisional prothesis."
In this case, this is the citation:
Marconcini S, Denaro M, Cosola S, Gabriele M, Toti P, Mijiritsky E, Proietti A, Basolo F, Giammarinaro E, Covani U. Myofibroblast Gene Expression Profile after Tooth Extraction in the Rabbit. Materials (Basel). 2019 Nov 9;12(22):3697. doi: 10.3390/ma12223697.
Response 7: The correction in the introduction was made. (page 3; lines 111-166)
8. DISCUSSION: You have to highligh the limitations of the study and possible future improvments. It is important for the Scientific Soundness of your paper.
Response 8: The correction in the discussion was made. (page 12, lines 380-395)
Best regards
